# A benchmark study of dioxygen complexes based on coupled cluster and density functional theory

Marcel Swart[a,b,*]

a) ICREA, Pg. Lluís Companys 23, 08010 Barcelona, Spain

b) IQCC and Dept. Chem., Univ. Girona, c/M.A. Capmany 69, 17003 Girona, Spain

marcel.swart@icrea.cat

## Abstract

A set of five compounds containing peroxo, superoxo or bis-μ-oxo moieties has been studied in the gas phase using CCSD(T)/aug-cc-pVQZ, in combination with Goodson's continued fraction approach. The corresponding analytical frequencies corroborate assignments of bands from experiments, and thus provide a consistent set of reference data that can be used for benchmarking a range of density functional approximations. A total of 75 density functionals have been checked for the bonds, peroxo/superoxo bonds, angles, vibrational frequencies and electronic energies. There is not one density functional that performs equally well for all of these properties, not even within one class of density functionals.

## Keywords

Density functional theory – Coupled cluster – Computational chemistry – Dioxygen compounds.

## Introduction

Dioxygen can coordinate to first-row transition metals (with different oxidation and spin states) with several protonation states, and hapticity, leading to e.g. peroxo, superoxo, hydroperoxo or bis-μ-oxo species (see Scheme 1). Although crystal structures exist for all of these species,[1,2] the corresponding complexes are too large for treatment by accurate wavefunction methods like CCSD(T) with large basis sets (aug-cc-pVQZ or better). This prevents the systematic validation[3–]

[8] of more efficient yet less accurate quantum-chemistry methods, e.g. based on density functional theory,[9] leading to uncertainty about the reliability and appropriateness of different density functionals[10,11] for this metal-dioxygen chemistry.

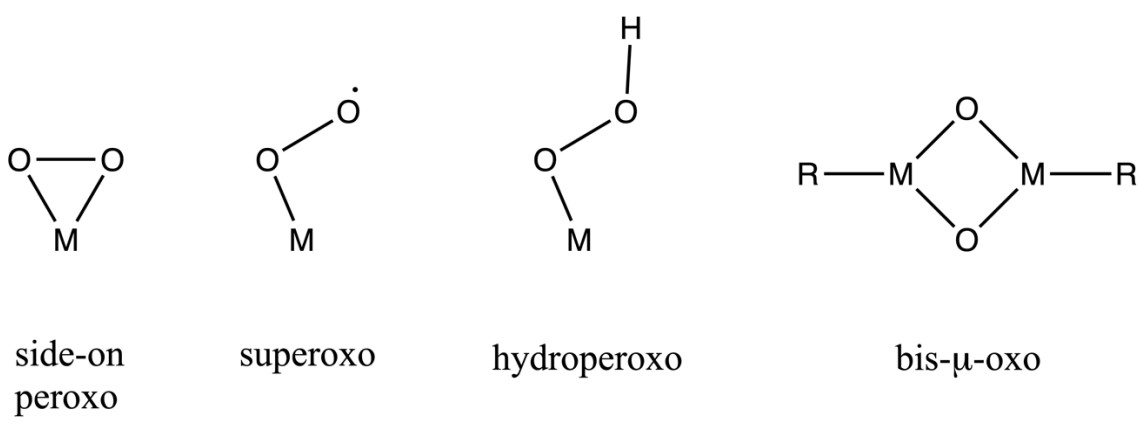

**Scheme 1**. Typical coordination of dioxygen towards (transition-)metal centers

Fortunately, in the early 2000s a series of papers by Schnöckel[12,13] and co-workers reported the detection of peroxo and bis-μ-oxo complexes of aluminium (see Scheme 2), with vibrational spectroscopic data available for characterization. Furthermore, Tremblay and Roy characterized silicon trioxide compounds spectroscopically.[14] Since these complexes are small (maximum six atoms), and highly symmetric, it is possible to study them by high-level wavefunction methods with sufficiently large basis sets. Hence, here I used CCSD(T)[15] with the aug-cc-pVQZ basis sets for complexes **1**-**5** (Scheme 2) to optimize their structure and compute the corresponding vibrational frequencies. These data can then serve as reference data for the development of new density functionals. Here, the results were used as reference data for the benchmarking of density functional approximations, to explore how well the latter reproduce the metal-O and O-O distances, the angles, the vibrational frequencies and the corresponding electronic energies.

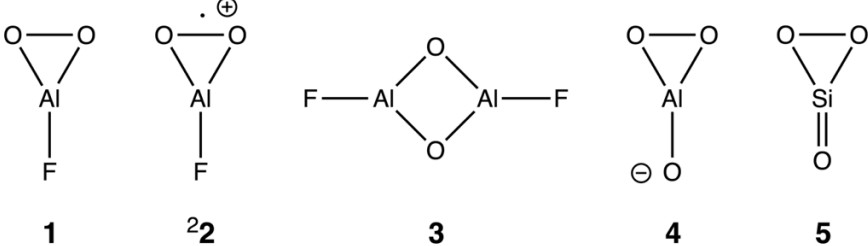

**Scheme 2**. Al/Si complexes studied in this paper. Experimental vibrational spectra are available for **1**,[12] **3**,[13] and for **5**.[14]

**Results and discussion**

Coupled cluster results. The experimentally observed vibrational data[12–14] for **1**, **3**, **5** are reported in Table 1, together with the CCSD(T)/aug-cc-pVQZ values. Also included are two related compounds, **2** and **4**, for which only the coupled cluster data are available. In general, there is a very good agreement between the experimentally observed peaks and the coupled cluster data. For instance, for $[FAlO_2]^0$ the peaks were observed experimentally at 1077 and 782 cm$^{-1}$, which match the harmonic coupled cluster frequencies of 1102 and 797 cm$^{-1}$; note that the difference between experiment and theory (2-3%) is most likely due to anharmonicity. On the other hand, for $[FAl(\mu\text{-}O)_2AlF]^0$ and $[OSi(O_2)]^0$ the differences between theory and experiment are smaller (ca. 1% or less). Whether this is due to changes in the effect of anharmonicity is unclear, and falls outside the scope of the current work (in the near future we will return to this issue).

Not all of the computed frequencies were observed experimentally, either due to too low intensity or not being able to assign these specifically to one of the manifolds of possible compounds that might be present in the experiments. For instance, in the reaction of AlF with $O_2$ the authors observed (at least) three species: **1**, **3** and a third one involving four oxygens (assigned[12] as a di-superoxo species $[FAl(O_2{}^{\bullet-})_2]^0$). Based on isotope effects when treating all of these species with either $^{16}O_2$, $^{18}O_2$ or a mixture of these, they were able to assign most of the observed peaks to one of these three species.

**Table 1**. Vibrational data (cm$^{-1}$) for dioxygen complexes[a]

| 1 | | 2 | 3 | | 4 | 5 | |
|---|---|---|---|---|---|---|---|
| exp.[b] | CCSD(T)[c] | CCSD(T)[c] | exp.[d] | CCSD(T)[c] | CCSD(T)[c] | exp.[e] | CCSD(T)[c] |
| 1077.3 | 1102.3 (160.6) | 1381.3 (3204.3) | 1015.0 | 1014.5 (0.0) | 1100.4 (82.8) | 1363.5 | 1382.2 (121.2) |
| 781.8 | 797.1 (26.7) | 1112.6 (189.8) | 942.8 | 966.1 (544.4) | 748.1 (25.5) | 877.1 | 887.6 (5.0) |
| - | 720.5 (8.8) | 1000.1 (98.6) | 812.9 | 821.4 (312.4) | 673.3 (27.2) | 855.3 | 860.4 (17.5) |
| - | 476.0 (13.0) | 612.1 (2.5) | - | 749.3 (0.0) | 544.0 (51.0) | - | 511.3 (16.9) |
| - | 228.5 (122.1) | 179.7 (199.1) | - | 690.5 (0.0) | 270.1 (33.2) | 292.0 | 307.2 (80.7) |
| - | 219.8 (71.5) | 173.5 (217.3) | - | 665.1 (32.1) | 256.8 (11.9) | 287.8 | 296.4 (53.0) |
| | | | - | 404.5 (0.0) | | | |
| | | | - | 400.7 (184.6) | | | |
| | | | - | 243.7 (0.0) | | | |
| | | | - | 225.2 (0.0) | | | |
| | | | - | 171.1 (30.5) | | | |
| | | | - | 94.5 (12.6) | | | |

a) shown in (parentheses) are the computed IR intensities (km·mol$^{-1}$); b) ref. [12]; c) CCSD(T)/aug-cc-pVQZ; d) ref. [13]; e) ref. [14]

The normal modes for $[FAlO_2]^0$ from CCSD(T)/aug-cc-pVQZ consist of an Al-F stretch at 1102 cm$^{-1}$, the anti-symmetric (797 cm$^{-1}$) and symmetric (721 cm$^{-1}$) Al-O$_2$ vibrations, the O-O vibration (476 cm$^{-1}$) and the out-of-plane ($\delta$-oop, 229 cm$^{-1}$) and in-plane ($\delta$-ip, 220 cm$^{-1}$) $\delta$(FAlOO) distortion. Similar modes are found for $[OAlO_2]^0$: $\nu$(Al-O) 1100 cm$^{-1}$, $\nu_{sym}$(Al-O$_2$) 748 cm$^{-1}$, $\nu_{asym}$(Al-O$_2$) 673 cm$^{-1}$, $\nu$(O-O) 544 cm$^{-1}$, $\nu$($\delta$-oop) 270 cm$^{-1}$, $\nu$($\delta$-ip) 257 cm$^{-1}$, and for $[OSi(O_2)]^0$: $\nu$(Si-O) 1382 cm$^{-1}$, $\nu_{sym}$(Si-O$_2$) 888 cm$^{-1}$, $\nu_{asym}$(Si-O$_2$) 860 cm$^{-1}$, $\nu$(O-O) 511 cm$^{-1}$, $\nu$($\delta$-oop) 307 cm$^{-1}$, $\nu$($\delta$-ip) 296 cm$^{-1}$. Note that in these latter cases the symmetric Al/Si-O$_2$ stretch shows a higher frequency than the asymmetric stretch, unlike the case of $[FAlO_2]^0$. Instead, for $^2[FAlO_2]^{\bullet+}$ the normal modes are shifted drastically: $\nu_{asym}$(Al-O$_2{}^{\bullet}$) 1381 cm$^{-1}$, $\nu$(Al-O) 1113 cm$^{-1}$, $\nu$(O-O)$^{\bullet}$ 1000 cm$^{-1}$, $\nu_{sym}$(Al-O$_2{}^{\bullet}$) 612 cm$^{-1}$, $\nu$($\delta$-ip) 180 cm$^{-1}$, $\nu$($\delta$-oop) 174 cm$^{-1}$. Finally, for $[FAl(\mu\text{-}O)_2AlF]^0$ the Al-F stretches are found at 1015 cm$^{-1}$ (sym) and 966 cm$^{-1}$ (asym), the Al-O$_2$-Al stretches at 691 cm$^{-1}$ (asym) and 665 cm$^{-1}$ (sym), and further modes involving the diamond-core breathing mode (749 cm$^{-1}$), FAl-AlF stretch (405 cm$^{-1}$), in-plane Al$_2$ vs. O$_2$ (821 cm$^{-1}$), out-of-plane Al$_2$ vs. O$_2$ (401 cm$^{-1}$), rocking (244 cm$^{-1}$) vs. flying (171 cm$^{-1}$) vs. out-of-plane (95 cm$^{-1}$) motions of Al$_2$O$_2$ vs. F$_2$,

and the anti-symmetric out-of-plane wobble of $Al_2F_2$ (225 cm$^{-1}$). Note that gif-movie files for all modes are available in Supporting Information.

**Table 2**. Geometric variables obtained at CCSD(T)/aug-cc-pVQZ

| bond | (Å) | angle | (°) |
|---|---|---|---|
| **1** | | | |
| F-Al | 1.62978 | F-Al-O | 150.286 |
| Al-O($O_2$) | 1.69710 | O-Al-O[a] | 59.428 |
| O-O | 1.68239 | | |
| **2** | | | |
| F-Al | 1.59587 | F-Al-O | 156.925 |
| Al-O($O_2$) | 1.78096 | O-Al-O[a] | 46.150 |
| O-O | 1.39605 | | |
| **3** | | | |
| F-Al | 1.63660 | F-Al-O | 133.328 |
| Al-O | 1.73798 | Al-O-Al | 86.655 |
| Al-Al[a] | 2.38510 | | |
| **4** | | | |
| O-Al | 1.63751 | F-Al-O | 152.282 |
| Al-O($O_2$) | 1.74850 | O-Al-O[a] | 55.437 |
| O-O | 1.62655 | | |
| **5** | | | |
| O-Si | 1.50991 | O-Al-O($O_2$) | 148.864 |
| Si-O($O_2$) | 1.62299 | O($O_2$)-Al-O($O_2$)[a] | 62.271 |
| O-O | 1.67839 | | |

a) not used for the deviations of density functional results (use of symmetry makes this value dependent on the other angle)

The geometric variables (bonds, angles) from the coupled cluster calculations are reported in Table 2. Based on the revised covalent radii by Alvarez and co-workers from 2008,[16] one would expect Al-F bonds of 1.78 Å, Al-O bonds of 1.87 Å, and Si-O bonds of 1.77 Å. Here, we observe Al-F, Al-O and Si-O bonds that are significantly shorter (0.15-0.20 Å). However, they are consistent with previous computational studies at lower levels of theory.[13,17,18] Furthermore, the coupled cluster

method with such a large basis set is often considered the 'gold standard' of computational chemistry. Indeed, the strong coherence between the computed and observed IR frequencies reinforce this notion. For this reason, the coupled cluster data seem to provide a good reference set to be able to compare density functional approximations with.

Density functional results. A total of 75 density functional approximations (DFAs) were tested against the coupled cluster reference data. Note that the same conditions were used as were done for coupled cluster: using the aug-cc-pVQZ basis in the gas phase and without relativistic corrections. Hence, I looked at the bond lengths and angles (see Table 2), frequencies, and difference with the electronic energy of Goodson's approach.[19] The latter check is merely informative, since DFT and coupled cluster approach the solution of the Schrödinger equation in a different manner. Density functional approaches modify the Hamiltonian, while the coupled cluster calculations improve the wavefunction within Schrödinger's equation. Nevertheless, in the end both approaches should result (at some point in the future) at the same total exact energy, and it is interesting to check how far we have come.

There were four DFAs (PWPB95, mPWB1K, mPW1B95, PW6B95) that showed SCF problems during one or more of the optimizations and frequency calculations, which in all cases involved the $^2[FAlO_2]^{\bullet+}$ system. Additionally, mPW1B95 showed problems as well for $[FAlO_2]^0$ and $[FAl(\mu\text{-}O)_2AlF]^0$. These problems were caused by the numerical instability of these functionals, which was solved by increasing the integration grid to (770,123) for the singlet systems. Further modifications were needed for the unrestricted $^2[FAlO_2]^{\bullet+}$ system: changing the SCF procedure to EDIIS and loosening the convergence criteria for energy ($10^{-7}$ Hartree) and density ($10^{-5}$ atomic units) solved the issues there as well.

The mean absolute deviation (MAD) from the coupled cluster reference data for *all* bonds (Table S1) is smallest for the MS1/MS2 metagga functionals (MAD values 0.0063/0.0066 Å), followed by TPSS (0.0080 Å), the MGGA-hybrid TPSSh (0.0092 Å), revised TPSS (0.0094 Å) and the

B97-D2 GGA (0.0096 Å). The best hybrid functional is B3LYP with a MAD value of 0.0118 Å, PW92 is the best LDA functional (0.0192 Å), HSE06 the best range-separated hybrid functional (0.0233 Å), B2PLYP the best double-hybrid functional (0.0356 Å) and PBEh-3c the best composite functional (0.0448 Å). The lower end of the table is filled with more exotic functionals like revSCAN0 or MVSh, but also includes double-hybrid functionals like DSD-PBEP86 (0.048 Å) or DSD-BLYP (0.0496 Å).

If we now specifically focus on the peroxo/superoxo O-O bonds alone (Table S2), a different picture emerges. B97-D2 remains performing well (0.0099 Å), followed closely by TPSS (0.0100 Å) and MS2 (0.0101 Å). The best MGGA-hybrid, hybrid, LDA and range-separated hybrid functionals are again TPSSh, B3LYP, PW92 and HSE06, but with MAD values (0.0215 Å, 0.0273 Å, 0.0411 Å, and 0.0518 Å, respectively) that are more than twice as large as for all bonds. The same holds for B2PLYP, which remains the best-performing double-hybrid functional, but with a MAD value for the peroxo/superoxo bonds of 0.0691 Å.

Most of the density functionals are able to predict well the angles (Table S3), with MAD values of less than 1°, but now with other best-performing functionals for most of the classes: BLYP for GGAs (MAD 0.208°), M06-L for MGGAs (MAD 0.282°), B3LYP remains best hybrid functional (MAD 0.368°), MS2h is the best MGGA-hybrid functional (MAD 0.378°), CAM-B3LYP the best range-separated hybrid functional (MAD 0.691°), PW92 best LDA functional (MAD 0.715°) and PTPSS the best double hybrid (MAD 0.855°).

The picture changes completely when looking at the vibrational frequencies (Table S4). Here the hybrid functionals in all the different guises are observed at the upper half of the list, with the B97-1 hybrid functional performing best (MAD 28.6 cm$^{-1}$), followed closely by B97 (28.7 cm$^{-1}$). The best MGGA-hybrid functional is now rev-TPSSh (MAD 30.0 cm$^{-1}$), M06-L is the best metagga functional (MAD 31.5 cm$^{-1}$), HSE06 the best range-separated hybrid functional (MAD 32.1 cm$^{-1}$), OPBE the best GGA functional (MAD 32.6 cm$^{-1}$), and PWPB95 the best double-hybrid functional

(MAD 42.1 cm$^{-1}$). Most of the list of 75 DFAs show MAD values for the vibrational frequencies that remain below 40 cm$^{-1}$, and 49 are within 10 cm$^{-1}$ from the best performing one.

Finally, out of curiosity I also checked how close are the DFA electronic energies to the cc-cf coupled cluster reference values. The double-hybrid functionals are among the best ones for this, with seven of them found in the top 10. Surprisingly, the three density functionals that come closest to coupled cluster are not among the top ones for the other four properties. The PKZB metagga functional has a MAD value from the cc-cf values of only 0.3 eV, followed by PBEh-3c (1.2 eV) and the DSD-BLYP double hybrid functional (2.3 eV). The latter two are among the poorest 15 functionals for the bonds, angles and frequencies, while for PKZB the situation is mixed: it is among the top 25 for angles and all bonds, among the top 10 for the peroxo/superoxo bonds, but among the lowest 25 for frequencies. For the rest there does not seem to be a clear trend between the class of functional, and the performance for these electronic energies (except for the double-hybrids, *vide supra*). For instance, the BLYP GGA (MAD 23.3 eV) is closer to coupled cluster than B3LYP (MAD 24.0 eV), but even closer are range-separated CAM-B3LYP (MAD 21.5 eV) and B1LYP (MAD 21.5 eV). However, the PBE and OPBE GGA functionals are even better, with MAD values of 12.0 eV and 17.7 eV, respectively; the corresponding PBE1PBE hybrid functional is almost as close as PBE with a MAD value of 12.1 eV. The revPBE (MAD 19.4 eV), RPBE (MAD 22.0 eV) and PBEsol (MAD 26.2 eV) are significantly farther away from the coupled cluster electronic energies than the non-empirical PBE.

The comparison of the LDA functionals adds some interesting results. The Perdew-Wang PW92 representation is for most properties the most accurate, with the VWN5 of Vosko-Wilk-Nusair giving almost identical results, and the VWN-RPA representation lagging both of them. However, this is reversed in the comparison of the electronic energies, where VWN-RPA gives a MAD value of 36.2 eV, while VWN5 and PW92 show drastically larger differences with 59.2 eV and 59.6 eV, respectively.

Overall assessment. Based on the MAD values for the five different properties (Table S1-S5), I also made an average assessment of the different density functionals. This was done by taking for each DFA the product of the MAD values for the different properties (PROD, shown in Table S6). The top 10 of this final list consists only of GGA and MGGA functionals, with PKZB at the top (PROD 0.001), followed by BP86 (PROD 0.003) and PBE (PROD 0.009). The best functionals for each class of DFAs are: PKZB (MGGA, PROD 0.001), BP86 (GGA, PROD 0.003), TPSSh (MGGA-hybrid, PROD 0.057), rev-B3LYP (hybrid, PROD 0.082), PBEh-3c (composite, PROD 0.219), HSE06 (range-separated hybrid, PROD 0.372), PTPSS (double hybrid, PROD 0.556) and VWN(RPA) (LDA, 1.001).

The average PROD value for all 75 functionals is rather high (5.292), but this is distorted by the two highest values for BHandH (31.296) and HF-3c (301.874). Instead, by taking the product of the averages for each of the five properties (Table 3), one obtains a PROD value of 0.8083.

**Table 3**. Statistics for deviations by DFAs (Table S1-S6) from CCSD(T)/aug-cc-pVQZ data

|  | min. MAD | max. MAD | average |
| --- | --- | --- | --- |
| all bonds (Å) | 0.0063 | 0.1291 | 0.0267 |
| dioxygen bonds (Å) | 0.0099 | 0.2357 | 0.0493 |
| angles (°) | 0.2082 | 2.857 | 0.7666 |
| frequencies (cm$^{-1}$) | 28.582 | 136.732 | 38.384 |
| electronic energies (eV) | 0.2875 | 59.5644 | 20.869 |
| product | 0.0012 | 301.8744 | 5.2921 |

**Conclusions**

A new set of small dioxygen compounds is presented, for which high-level ab initio calculations were feasible at the CCSD(T)/aug-cc-pVQZ level. The resulting data could be used as reference in the development of new density functionals, or used to benchmark existing density functionals.

The latter is done here, for a total of 75 density functionals coming from eight classes (LDA, GGA, MGGA, hybrid, MGGA hybrid, range-separated hybrid, double hybrid and composites). In general, the density functionals are remarkably accurate for bonds and angles, with average deviations of 0.027 Å for *all* bonds, 0.77° for angles; dioxygen bonds are more difficult to get right, with an average deviation from the coupled cluster of 0.049 Å, almost twice as large as the deviation for all bonds. The frequencies from coupled cluster calculations are off by ca. 30-40 cm$^{-1}$, and there is still a long way to go before density functional theory and coupled cluster theory converge to the same exact energy: for the systems studied here on average a deviation of 20.9 eV was observed.

**Computational details**

Coupled cluster: All CCSD(T) calculations with the aug-cc-pVQZ[20,21] basis set were carried out with CFOUR (version 2.0/2.1),[22,23] in parallel where needed,[24] using unrestricted Hartree-Fock for open-shell systems. Note that a bugfix (Dec. 2021, see SI) was needed for being able to run in parallel with unrestricted coupled cluster protocols. The frozen core approach was used within the solving of the coupled cluster equations. The geometry optimization typically had SCF and coupled-cluster convergence criteria of $1.0 \cdot 10^{-10}$, geometry converge of $1.0 \cdot 10^{-8}$ atomic units, leading to electronic energies accurate to at least $1.0 \cdot 10^{-8}$ Hartree. The two-electron integrals were in all cases treated with the AOBASIS formalism to reduce disk space, in conjunction with the ECC program. Goodson's continued fraction approach[19] was used to estimate a closer approximation of the full-CI electronic energy (cc-cf), based on the CCSD and CCSD(T) energies. As shown by Schröder and co-workers,[25] Goodson's approach does not directly extrapolate towards the full-CI limit, but rather to the next excitation level in the coupled cluster series, which in this case would be CCSDTQ.

Density functional theory: All density functional calculations were calculated with the psi4 program (release 1.6.1).[26–28] For optimizations, the most tight criteria available in the

program were used (*gradient max 2.0·10⁻⁶ a.u., RMS force 1.0·10⁻⁶ a.u., max displacement*
*6.0·10⁻⁶ a.u., rms displacement 4.0·10⁻⁶ a.u.*), with typically a grid based on 590 spherical points and 99 radial points. The doublet states (**2**) were treated using the unrestricted Kohn-Sham formalism. Similar to coupled cluster, the DFA calculations were performed in the gas-phase, without relativistic corrections, and using the aug-cc-pVQZ basis set. Some of the results for density functional approximations were obtained through the LibXC[29] library, either the one available directly in psi4, or compiled separately and linked to it. We have studied DFAs of different classes: local density approximation (VWN(RPA),[30] VWN5[30] and PW92[31]), generalized gradient approximations (B97-d,[32] BLYP,[33,34] BP86-PZ81,[33,35,36] BP86-VWN,[30,33,35] OPBE,[37–39] PBE,[39] PBEsol,[40] PW91,[41] revPBE,[42] RPBE,[43] S12g,[44] SOGGA,[45] XLYP[46]), meta-GGA approximations (M06-L,[47] MS1,[48] MS2,[48] MVS,[49] PKZB,[50] revSCAN,[51] revTPSS,[52] SCAN,[53] TPSS[54]), hybrid functionals (B1LYP,[55] B1PW91,[55] B3LYP,[33,34,56] B3PW91,[33,41,57] B97,[58] B97-1,[59] B97-2,[60] B97-3,[61] B97-k,[62] BHandH,[33,57] BHandHLYP,[33,34,57] HJS-PBE,[63] LDA1LDA,[64] mPW1K,[65] mPW1PW,[66] mPWB1K,[67] O3LYP,[34,38,68] PBE1PBE,[39,69] revB3LYP,[70] revPBE1PBE,[39,42,69] SOGGA11-X,[71] X3LYP[46]), hybrid meta-GGA (BB1K,[72] BMK,[62] M06,[73] M06-2X,[73] M08-HX,[74] M11,[75] MS2h,[48] MVSh,[49] MN15,[76] mPW1B95,[67] PW6B95,[77] revSCAN1SCAN,[51] revTPSSh,[78] SCAN1SCAN,[79] TPSSh[54,80]), range-separated hybrids (CAM-B3LYP,[81] HSE06,[82–84] ωB97X-D[85,86]), double hybrids (B2GPPLYP,[87] B2PLYP,[88] DSD-BLYP,[89] DSD-PBEP86,[89] PBE1PBE-DH,[90] PTPSS,[91] PWPB95[91]) and composite methods (HF-3c,[92,93] PBEh-3c[92,93]).

## Acknowledgments

The author would like to thank AEI/MCIN (CTQ2017-87392-P, PID2020-114548GB-I00) and GenCat (2021 SGR 00487) for funding.

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
