# Peer review of "A benchmark study of dioxygen complexes based on coupled cluster and density functional theory"

_SciPost Chemistry_

## Round 1 · Referee Report · Anonymous (Referee 1) · 2023-9-8

Strengths

  1. The study reports on an interesting topic, and rather systematic benchmarks are (sadly) seldom published as they take a lot of time to do, and are sometimes difficult to get accepted in journals ("what is the new chemical insight?"). This methodological work should be encouraged.

  2. The reporting is done in a rather concise way, with data published as supporting information. This makes for an easy read, not getting down too much into the details in the main paper.

  3. The methodology is reasonably well described, and the comparison with experimental data allows this to go beyond a simple "X vs Y" computational study.

Weaknesses

  1. The author has gather a lot of data, but I do not think they exploit it sufficiently. In particular, there is no real graphic representation or statistical analysis of the trends. I encourage the author to add some further analysis, showing distributions of values, correlations between different functionals (within the same family and outside), etc. This would allow to give a more quantitative conclusion than "there is no single best performing functional" (which is somewhat expected). Maybe synthesizing the performance of every function on each metric (geometry, frequencies, etc.) with a "figure of merit"? I don't want to be too prescriptive, but I think further statistical analysis would strongly improve the paper. Right now, it lacks a bit in the depth of analysis and conclusions.

1bis. Please provide graphical representation of the data, so it is easier to read than tables.

  1. The data gathered is published only inside a PDF file, with page breaks, page numbers, comments. It is not machine readable, and very far from the FAIR principles for research data publication. Please publish the data sets gathered in an open, machine readable format (CSV, JSON, etc). This would allow other groups to reuse it easily, make their own analysis, complement the work and have a lot more value to the community.

  2. For full reproducibility of the work for others, it would be necessary to publish the input files for the calculations. Even though the methodology is described, every calculation has tons of small "hidden parameters" which can be important for reproducing existing work. Input files, optimized geometries should be published (again, as machine-readable files in a ZIP archive or an online repository like Zenodo). Otherwise, the promise of the author ("The resulting data could be used as reference in the development of new density functionals, or used to benchmark existing density functionals.") is somewhat empty.

Report

I think this study can be improved and, at a later stage, accepted for publication. With some extension, I expect it will meet the criteria for publication.

Requested changes

See above.

---

## Round 1 · Referee Report · Anonymous (Referee 2) · 2023-9-9

Report

In this manuscript, Swart studied the performance of 75 density functionals against CCSD(T) for 5 compounds. However, due to the fatal mistake identified (see questions 5) and the contents, I am concerned that it may not meet the publication requirements.

  1. The author used the CFOUR package to perform the first and second derivative calculations of coupled cluster with singles and doubles and perturbative triples (CCSD(T)) for geometry optimization and Hessian. However, in the paper, the author only cites the classic quantum chemistry textbook, ref. [15], instead of detailed CCSD(T) works. As far as I know, the book only contains chapters about the first derivative for closed-shell CCSD. However, in the paper, the author used analytical gradients and Hessian of CCSD(T) for both closed- and open-shell references. The correct references should be cited, e.g., J.D. Watts, J. Gauss, and R.J. Bartlett, Open-shell analytical energy gradients for triple excitation many-body, coupled-cluster methods: MBPT(4), CCSD+T(CCSD), CCSD(T),and QCISD(T), Chem. Phys. Lett. 200, 1-7 (1992) J. Gauss and J.F. Stanton, Analytic CCSD(T) second derivatives, Chem. Phys. Letters 276, 70-77 (1997) P.G. Szalay, J. Gauss, and J.F. Stanton, Analytic UHF-CCSD(T) second derivatives: implementation and application to the calculation of the vibration-rotation interaction constants of NCO and NCS, Theor. Chim. Acta 100, 5-11 (1998) Moreover, in the paper, the author mentioned that the 'ECC program' is used. Please clarify whether 'ECC' refers to a conventional UCCSD(T) or a spin-adapted UCCSD(T).

  2. It is essential to explicitly state the charge and spin (multiplicity) used in the calculations for all five compounds under investigation. Regarding the three compounds with available experimental data, could the authors clarify whether the spin information is also derived from the experiments? To enhance the understanding of the remaining two compounds without experimental data, I recommend that the author conducts CCSD(T) calculations with different multiplicities. This will help in distinguishing between ground and excited states in the study.

  3. The paper and its supplementary information mention an issue when parallelizing the calculation of open-shell CCSD(T) energies. Upon reviewing reference 24, it appears that the parallel implementation for open-shell second derivatives has not been developed. Could the authors provide clarification on this matter? To my understanding, analytical gradient and Hessian algorithms tend to be more complex than energy calculations. I'm curious if the parallel versions of the first and second-order derivative algorithms also encounter issues at the open-shell CCSD(T) level. Could the authors provide clarification on the presence of any such bugs in these derivative algorithms?

  4. Open-shell systems often exhibit spin contaminations in their UCCSD(T) wave functions. It is recommended that the authors report the value of <S^2> for the UCCSD wave function. To address spin contaminations, Szalay and colleagues introduced a spin-restricted method for open-shell systems, as described in their work (J. Chem. Phys. 107, 9028–9038 (1997)). If the calculated value of <S^2> for the open-shell UCCSD differs significantly from the ideal value, it is advisable to consider the use of a spin-adapted UCCSD(T) approach.

  5. If I comprehend this correctly, in addition to exploring various second-order properties, the author has compared absolute energies obtained from different DFT methods with those from CCSD(T) or cc-cf calculations, which, in my view, appears illogical. Absolute energies derived from various single-reference or multi-reference CI methods could be valuable, considering the variation principle. However, the absolute MPn or CC energies should be approached with caution unless their wave functions have been properly normalized. To the best of my knowledge, comparing absolute energies obtained from DFT calculations is discouraged, and the focus should be on relative energies. Consequently, the results presented in Table S5 may not provide meaningful insights. Rather than absolute energies, it would be more meaningful if the author computed atomization energies (relative energies) using both CCSD(T) and DFT methods. Given that absolute energies are not considered meaningful in this context, it follows that the extrapolated FCI energies obtained via the Goodson method may also lack significance. However, it would still be valuable for the paper to report atomization energies based on the extrapolated CI energies

  6. In my view, instead of starting with benchmarking the first (geometry) and second-order derivatives of energies for oxo compounds, the author should consider commencing with energy-related benchmarks. Atomization energies would be a suitable choice for such a benchmark.

  7. In the abstract, the author mentions, 'There is not one density functional that performs equally well for all of these properties, not even within one class of density functionals.' However, I couldn't find any analysis comparing different types of DFT, such as LDA or GGA. It would be beneficial to identify at least one type of DFT that demonstrates relatively better performance in terms of predicting geometry or vibrational frequencies. Without such analysis, the paper lacks valuable information beyond the CCSD(T) results.

---

## Round 1 · Referee Report · Anonymous (Referee 3) · 2023-9-10

Report

The work by Swart reports a benchmark study of 75 density functionals against CCSD(T) for five Al/Si-oxygen complexes. For bond length, bond angles, vibrational frequencies and electronic energies tested in this study, it was concluded that the none of DFT functionals can performs equally well for all of these properties.
(1) I am not sure if the electronic energies are comparable between CCSD(T) and different functionals and how meaningful to do this test?
(2) I think it is generally recognized that DFT functionals performs well for the ground-state geometries, especially for the complexes composed of main group elements, as selected in this study.
(3) I feel it would be more meaningful to evaluate the performance of DFT in terms of transition state geometry or kinetic barriers, which could be more instructional.

---

## Round 1 · Referee Report · Anonymous (Referee 4) · 2023-9-12

Strengths

  1. A new set of benchmark data for dioxygen complexes is reported.
  2. A large number of density functional approximations are tested against the data.
  3. The supporting information contains enough "input file" information to easily reproduce the coupled cluster calculations.
  4. GIF files are provided as supporting information to visualise the vibrational normal modes.

Weaknesses

  1. It is not clear how useful the comparison of absolute total energies is.
  2. There is a known deficiency in the basis set selected and it is unclear what impact this may have on the results.

Report

In producing a new benchmarking data set at a high-level of theory, this work has clear potential for follow up work, particularly in the development and testing of new density functional approximations.

In general it is clearly written with suitable abstract and introduction. The combination of the computational details and supporting information means that the work is easily reproducible and there are appropriate citations.

To further address the highlighted weaknesses: 1. It is not clear that the total energy comparison is useful, particularly as the coupled cluster method is non-variational. Some form of relative energies would seem more appropriate and presumably remove the large outlier results that are evident in Table 3. Total atomisation energies would be a possibility here and should be relatively easy to compute as an extension of the calculations already performed.

  1. The inclusion of "tight" d functions in the basis set is known to be important for second row elements such as Al and Si (see DOI: 10.1063/1.1367373 ) but not used in this investigation. For a benchmark study such as this, the impact of including such functions should be known.

I also note that the continued fraction method is not that well known outside of the high-accuracy thermochemistry community. I would recommend giving slightly more information on this, including the key equation, as part of the manuscript.

Requested changes

  1. Replace the comparison of total electronic energies with total atomisation energies.

  2. Calibrate the effect of including tight d functions in the basis for Al and Si. This could potentially be explored at a lower-zeta level [for example comparing aug-cc-pV(D+d)Z with aug-cc-pVDZ] to determine if it is significant for these particular systems.

  3. Briefly expand the description of the continued fraction method.

---

## Editorial Decision

resubmitted